# MoCap-guided Data Augmentation for 3D Pose Estimation in the Wild

**Grégory Rogez**       **Cordelia Schmid**

Inria Grenoble Rhône-Alpes, Laboratoire Jean Kuntzmann, France

## Abstract

This paper addresses the problem of 3D human pose estimation in the wild. A significant challenge is the lack of training data, i.e., 2D images of humans annotated with 3D poses. Such data is necessary to train state-of-the-art CNN architectures. Here, we propose a solution to generate a large set of photorealistic synthetic images of humans with 3D pose annotations. We introduce an image-based synthesis engine that artificially augments a dataset of real images with 2D human pose annotations using 3D Motion Capture (MoCap) data. Given a candidate 3D pose our algorithm selects for each joint an image whose 2D pose locally matches the projected 3D pose. The selected images are then combined to generate a new synthetic image by stitching local image patches in a kinematically constrained manner. The resulting images are used to train an end-to-end CNN for full-body 3D pose estimation. We cluster the training data into a large number of pose classes and tackle pose estimation as a K-way classification problem. Such an approach is viable only with large training sets such as ours. Our method outperforms the state of the art in terms of 3D pose estimation in controlled environments (Human3.6M) and shows promising results for in-the-wild images (LSP). This demonstrates that CNNs trained on artificial images generalize well to real images.

## 1   Introduction

Convolutionnal Neural Networks (CNN) have been very successful for many different tasks in computer vision. However, training these deep architectures requires large scale datasets which are not always available or easily collectable. This is particularly the case for 3D human pose estimation, for which an accurate annotation of 3D articulated poses in large collections of real images is non-trivial: annotating 2D images with 3D pose information is impractical [6] while large scale 3D pose capture is only available through marker-based systems in constrained environments [13]. The images captured in such conditions do not match well real environments. This has limited the development of end-to-end CNN architectures for in-the-wild 3D pose understanding.

Learning architectures usually augment existing training data by applying synthetic perturbations to the original images, e.g. jittering exemplars or applying more complex affine or perspective transformations [15]. Such data augmentation has proven to be a crucial stage, especially for training deep architectures. Recent work [14, 23, 34, 40] has introduced the use of data synthesis as a solution to train CNNs when only limited data is available. Synthesis can potentially provide infinite training data by rendering 3D CAD models from any camera viewpoint [23, 34, 40]. Fisher et al [8] generate a synthetic "Flying Chairs" dataset to learn optical flow with a CNN and show that networks trained on this unrealistic data still generalize very well to existing datasets. In the context of scene text recognition, Jaderberg et al. [14] trained solely on data produced by a synthetic text generation engine. In this case, the synthetic data is highly realistic and sufficient to replace real data. Although synthesis seems like an appealing solution, there often exists a large domain shift from synthetic to real data [23]. Integrating a human 3D model in a given background in a realistic way is not trivial. Rendering a collection of photo-realistic images (in terms of color, texture, context, shadow) that would cover the variations in pose, body shape, clothing and scenes is a challenging task.

Instead of rendering a human 3D model, we propose an image-based synthesis approach that makes use of Motion Capture (MoCap) data to augment an existing dataset of real images with 2D pose

annotations. Our system synthesizes a very large number of new in-the-wild images showing more pose configurations and, importantly, it provides the corresponding 3D pose annotations (see Fig. 1). For each candidate 3D pose in the MoCap library, our system combines several annotated images to generate a synthetic image of a human in this particular pose. This is achieved by "copy-pasting" the image information corresponding to each joint in a kinematically constrained manner. Given this large "in-the-wild" dataset, we implement an end-to-end CNN architecture for 3D pose estimation. Our approach first clusters the 3D poses into K pose classes. Then, a K-way CNN classifier is trained to return a distribution over probable pose classes given a bounding box around the human in the image. Our method outperforms state-of-the-art results in terms of 3D pose estimation in controlled environments and shows promising results on images captured "in-the-wild".

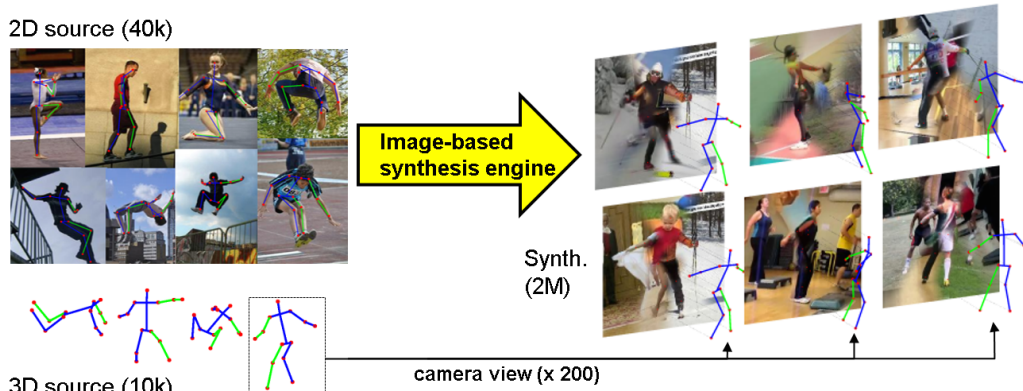

Figure 1: Image-based synthesis engine. Input: real images with manual annotation of 2D poses, and 3D poses captured with a Motion Capture (MoCap) system. Output: 220x220 synthetic images and associated 3D poses.

## 1.1 Related work

**3D human pose estimation in monocular images.** Recent approaches employ CNNs for 3D pose estimation in monocular images [20] or in videos [44]. Due to the lack of large scale training data, they are usually trained (and tested) on 3D MoCap data in constrained environments [20]. Pose understanding in natural images is usually limited to 2D pose estimation [7, 36, 37]. Recent work also tackles 3D pose understanding from 2D poses [2, 10]. Some approaches use as input the 2D joints automatically provided by a 2D pose detector [32, 38], while others jointly solve the 2D and 3D pose estimation [31, 43]. Most similar to ours is the approach of Iqbal et al. [42] who use a dual-source approach that combines 2D pose estimation with 3D pose retrieval. Our method uses the same two training sources, i.e., images with annotated 2D pose and 3D MoCap data. However, we combine both sources off-line to generate a large training set that is used to train an end-to-end CNN 3D pose classifier. This is shown to improve over [42], which can be explained by the fact that training is performed in an end-to-end fashion.

**Synthetic pose data.** A number of works have considered the use of synthetic data for human pose estimation. Synthetic data have been used for upper body [29], full-body silhouettes [1], hand-object interactions [28], full-body pose from depth [30] or egocentric RGB-D scenes [27]. Recently, Zuffi and Black [45] used a 3D mesh-model to sample synthetic exemplars and fit 3D scans. In [11], a scene-specific pedestrian detectors was learned without real data while [9] synthesized virtual samples with a generative model to enhance the classification performance of a discriminative model. In [12], pictures of 2D characters were animated by fitting and deforming a 3D mesh model. Later, [25] augmented labelled training images with small perturbations in a similar way. These methods require a perfect segmentation of the humans in the images. Park and Ramanan [22] synthesized hypothetical poses for tracking purposes by applying geometric transformations to the first frame of a video sequence. We also use image-based synthesis to generate images but our rendering engine combines image regions from several images to create images with associated 3D poses.

## 2 Image-based synthesis engine

At the heart of our approach is an image-based synthesis engine that artificially generates "in-the-wild" images with 3D pose annotations. Our method takes as input a dataset of real images with 2D

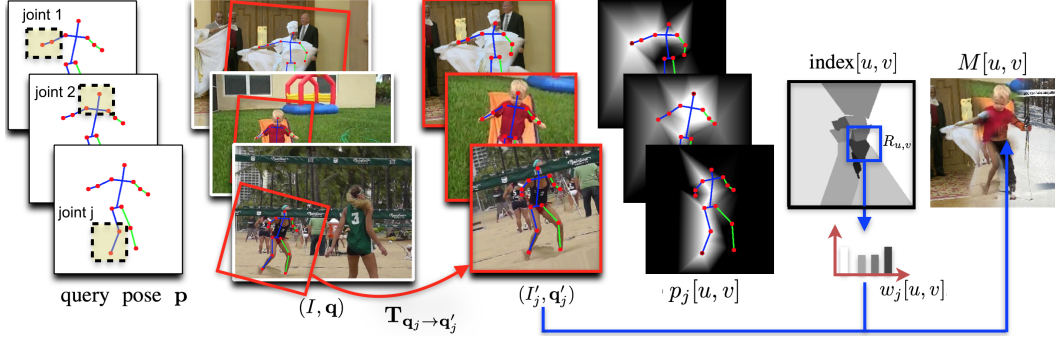

Figure 2: Synthesis engine. From left to right: for each joint $j$ of a 2D query pose $\mathbf{p}$ (centered in a $220 \times 220$ bounding box), we align all the annotated 2D poses w.r.t the limb and search for the best pose match, obtaining a list of $n$ matches $\{(I'_j, \mathbf{q}'_j), j = 1...n\}$ where $I'_j$ is obtained after transforming $I_j$ with $\mathbf{T}\mathbf{q}_j \to \mathbf{q}'_j$. For each retrieved pair, we compute a probability map $p_j[u, v]$. These $n$ maps are used to compute $\text{index}[u, v] \in \{1...n\}$, pointing to the image $I'_j$ that should be used for a particular pixel $(u, v)$. Finally, our blending algorithm computes each pixel value of the synthetic image $M[u, v]$ as the weighted sum over all aligned images $I'_j$, the weights being calculated using an histogram of indexes in a squared region $R_{u,v}$ around $(u, v)$.

annotations and a library of 3D Motion Capture (MoCap) data, and generates a large number of synthetic images with associated 3D poses (Fig. 1). We introduce an image-based rendering engine that augments the existing database of annotated images with a very large set of photorealistic images covering more body pose configurations than the original set. This is done by selecting and stitching image patches in a kinematically constrained manner using the MoCap 3D poses. Our synthesis process consists of two stages: a MoCap-guided mosaic construction stage that stitches image patches together and a pose-aware blending process that improves image quality and erases patch seams. These are discussed in the following subsections. Fig. 2 summarizes the overall process.

## 2.1 MoCap-guided image mosaicing

Given a 3D pose with $n$ joints $\mathbf{P} \in \mathbb{R}^{\mathbf{n} \times \mathbf{3}}$, and its projected 2D joints $\mathbf{p} = \{p_j, j = 1...n\}$ in a particular camera view, we want to find for each joint $j \in \{1...n\}$ an image whose annotated 2D pose presents a similar kinematic configuration around $j$. To do so, we define a distance function between 2 different 2D poses $\mathbf{p}$ and $\mathbf{q}$, conditioned on joint $j$ as:

$$D_j(\mathbf{p}, \mathbf{q}) = \sum_{k=1}^{n} d_{\text{E}}(p_k, q'_k) \tag{1}$$

where $d_{\text{E}}$ is the Euclidean distance. $\mathbf{q}'$ is the aligned version of $\mathbf{q}$ with respect to joint $j$ after applying a rigid transformation $\mathbf{T}_{\mathbf{q}_j \to \mathbf{q}'_j}$, which respects $q'_j = p_j$ and $q'_i = p_i$ , where $i$ is the farthest directly connected joint to $j$ in $\mathbf{p}$. This function $D_j$ measures the similarity between 2 joints by aligning and taking into account the entire poses. To increase the influence of neighboring joints, we weight the distances $d_{\text{E}}$ between each pair of joints $\{(p_k, q'_k), k = 1...n\}$ according to their distance to the query joint $j$ in both poses. Eq. 1 becomes:

$$D_j(\mathbf{p}, \mathbf{q}) = \sum_{k=1}^{n} (w_k^j(\mathbf{p}) + w_k^j(\mathbf{q})) \, d_{\text{E}}(p_k, q'_k) \tag{2}$$

where weight $w_k^j$ is inversely proportional to the distance between joint $k$ and the query joint $j$, i.e., $w_k^j(\mathbf{p}) = 1/d_{\text{E}}(p_k, p_j)$ and normalized so that $\sum_k w_k^j(\mathbf{p}) = 1$. For each joint $j$ of the query pose $\mathbf{p}$, we retrieve from our dataset $\mathbb{Q} = \{(I_1, \mathbf{q_1}) \dots (I_N, \mathbf{q}_N)\}$ of images and annotated 2D poses[1]:

$$\mathbf{q}_j = \text{argmin}_{\mathbf{q} \in \mathbb{Q}} D_j(\mathbf{p}, \mathbf{q}) \quad \forall j \in \{1...n\}. \tag{3}$$

We obtain a list of $n$ matches $\{(I'_j, \mathbf{q}'_j), j = 1...n\}$ where $I'_j$ is the cropped image obtained after transforming $I_j$ with $\mathbf{T}_{\mathbf{q}_j \to \mathbf{q}'_j}$. Note that a same pair $(I, \mathbf{q})$ can appear multiple times in the list of candidates, i.e., being a good match for several joints.

Finally, to render a new image, we need to select the candidate images $I_j'$ to be used for each pixel $(u, v)$. Instead of using regular patches, we compute a probability map $p_j[u, v]$ associated with each pair $(I_j', \mathbf{q}_j')$ based on local matches measured by $d_{\mathrm{E}}(p_k, q_k')$ in Eq. 1. To do so, we first apply a Delaunay triangulation to the set of 2D joints in $\{\mathbf{q}_j'\}$ obtaining a partition of the image into triangles, accordingly to the selected pose. Then, we assign the probability $p_j(q_k') = exp(-d_{\mathrm{E}}(p_k, q_k')^2/\sigma^2)$ to each vertex $q_k'$. We finally compute a probability map $p_j[u, v]$ by interpolating values from these vertices using barycentric interpolation inside each triangle. The resulting $n$ probability maps are concatenated and an index map $\text{index}[u, v] \in \{1...n\}$ can be computed as follows:

$$\text{index}[u, v] = \text{argmax}_{j \in \{1...n\}} \, p_j[u, v], \tag{4}$$

this map pointing to the training image $I_j'$ that should be used for each pixel $(u, v)$. A mosaic $M[u, v]$ can be generated by "copy-pasting" image information at pixel $(u, v)$ indicated by $\text{index}[u, v]$:

$$M[u, v] = I_{j*}'[u, v] \quad \text{with} \quad j^* = \text{index}[u, v]. \tag{5}$$

## 2.2 Pose-aware image blending

The mosaic $M[u, v]$ resulting from the previous stage presents significant artifacts at the boundaries between image regions. Smoothing is necessary to prevent the learning algorithm from interpreting these artifacts as discriminative pose-related features. We first experimented with off-the-shelf image filtering and alpha blending algorithms, but the results were not satisfactory. Instead, we propose a new pose-aware blending algorithm that maintains image information on the human body while erasing most of the stitching artifacts. For each pixel $(u, v)$, we select a surrounding squared region $R_{u,v}$ whose size varies with the distance of pixel $(u, v)$ to the pose: $R_{u,v}$ will be larger when far from the body and smaller nearby. Then, we evaluate how much each image $I_j'$ should contribute to the value of pixel $(u, v)$ by building an histogram of the image indexes inside the region $R_{u,v}$:

$$w_j[u, v] = \text{Hist}(\text{index}(R_{u,v})) \,\, \forall j \in \{1 \ldots n\}, \tag{6}$$

where the weights are normalized so that $\sum_j w_j[u, v] = 1$. The final mosaic $M[u, v]$ (see examples in Fig. 1) is then computed as the weighted sum over all aligned images:

$$M[u, v] = \sum_j w_j[u, v] I_j'[u, v]. \tag{7}$$

This procedure produces plausible images that are kinematically correct and locally photorealistic.

## 3 CNN for full-body 3D pose estimation

Human pose estimation has been addressed as a classification problem in the past [4, 21, 27, 26]. Here, the 3D pose space is partitioned into K clusters and a K-way classifier is trained to return a distribution over pose classes. Such a classification approach allows modeling multimodal outputs in ambiguous cases, and produces multiple hypothesis that can be rescored, e.g. using temporal information. Training such a classifier requires a reasonable amount of data per class which implies a well-defined and limited pose space (e.g. walking action) [26, 4], a large-scale synthetic dataset [27] or both [21]. Here, we introduce a CNN-based classification approach for full-body 3D pose estimation. Inspired by the DeepPose algorithm [37] where the AlexNet CNN architecture [19] is used for full-body 2D pose regression, we select the same architecture and adapt it to the task of 3D body pose classification. This is done by adapting the last fully-connected layer to output a distribution of scores over pose classes as illustrated in Fig. 3. Training such a classifier requires a large amount of training data that we generate using our image-based synthesis engine.

Given a library of MoCap data and a set of camera views, we synthesize for each 3D pose a $220 \times 220$ image. This size has proved to be adequate for full-body pose estimation [37]. The 3D poses are then aligned with respect to the camera center and translated to the center of the torso. In that way, we obtain orientated 3D poses that also contain the viewpoint information. We cluster the resulting 3D poses to define our classes which will correspond to groups of similar orientated 3D poses. We empirically found that K=5000 clusters was a sufficient number of clusters. For evaluation, we return the average 2D and 3D poses of the top scoring class.

To compare with [37], we also train a holistic pose regressor, which regresses to 2D and 3D poses (not only 2D). To do so, we concatenate the 3D coordinates expressed in meters normalized to the range $[-1, 1]$, with the 2D pose coordinates, also normalized in the range $[-1, 1]$ following [37].

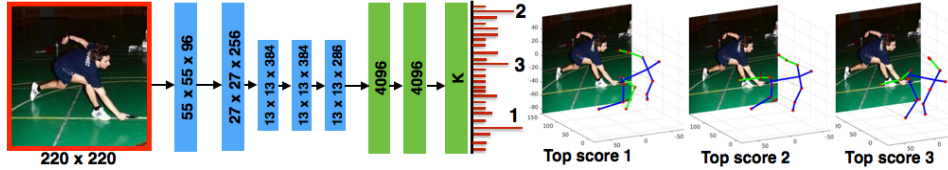

Figure 3: CNN-based pose classifier. We show the different layers with their corresponding dimensions, with convolutional layers depicted in blue and fully connected ones in green. The output is a distribution over K pose classes. Pose estimation is obtained by taking the highest score in this distribution. We show on the right the 3D poses for 3 highest scores.

## 4 Experiments

We address 3D pose estimation in the wild. However, there does not exist a dataset of real-world images with 3D annotations. We thus evaluate our method in two different settings using existing datasets: (1) we validate our 3D pose predictions using Human3.6M [13] which provides accurate 3D and 2D poses for 15 different actions captured in a controlled indoor environment; (2) we evaluate on Leeds Sport dataset (LSP)[16] that presents in-the-wild images together with full-body 2D pose annotations. We demonstrate competitive results with state-of-the-art methods for both of them.

Our image-based rendering engine requires two different training sources: 1) a 2D source of images with 2D pose annotations and 2) a MoCap 3D source. We consider two different datasets for each: for 3D poses we use the CMU Motion Capture Dataset[2] and the Human3.6M 3D poses [13], and for 2D pose annotations the MPII-LSP-extended dataset [24] and the Human3.6M 2D poses and images.

**MoCap 3D source.** The CMU Motion Capture dataset consists of 2500 sequences and a total of 140,000 3D poses. We align the 3D poses w.r.t. the torso and select a subset of 12,000 poses, ensuring that selected poses have at least one joint 5 cm apart. In that way, we densely populate our pose space and avoid repeating common poses (e.g. neutral standing or walking poses which are over-represented in the dataset). For each of the 12,000 original MoCap poses, we sample 180 random virtual views with azimuth angle spanning 360 degrees and elevation angles in the range $[-45, 45]$. We generate over 2 million pairs of 3D/2D pose configurations (articulated poses + camera position and angle). For Human3.6M, we randomly selected a subset of 190,000 orientated 3D poses, discarding similar poses, i.e., when the average Euclidean distance of the joints is less than 15mm as in [42].

**2D source.** For the training dataset of real images with 2D pose annotations, we use the MPII-LSP-extended [24] which is a concatenation of the extended LSP [17] and the MPII dataset [3]. Some of the poses were manually corrected as a non-negligible number of annotations are not accurate enough or completely wrong (eg., right-left inversions or bad ordering of the joints along a limb). We mirror the images to double the size of the training set, obtaining a total of 80,000 images with 2D pose annotations. For Human3.6M, we consider the 4 cameras and create a pool of 17,000 images and associated 2D poses that we also mirror. We ensure that most similar poses have at least one joint 5 cm apart in 3D.

### 4.1 Evaluation on Human3.6M Dataset (H3.6M)

To compare our results with very recent work in 3D pose estimation [42], we follow the protocol introduced in [18] and employed in [42]: we consider six subjects (S1, S5, S6, S7, S8 and S9) for training, use every $64^{th}$ frame of subject S11 for testing and evaluate the 3D pose error (mm) averaged over the 13 joints. We refer to this protocol by P1. As in [42], we consider a 3D pose error that measures accuracy of aligned pose by a rigid transformation but also report the absolute error.

We first evaluate the impact of our synthetic data on the performances for both the regressor and classifier. The results are reported in Tab. 1. We can observe that when considering few training images (17,000), the regressor clearly outperforms the classifier which, in turns, reaches better performances when trained on larger sets. This can be explained by the fact that the classification approach requires a sufficient amount of examples. We, then, compare results when training both regressor and classifier on the same 190,000 poses considering a) synthetic data generating from H3.6M, b) the real images corresponding to the 190,000 poses and c) the synthetic and real images

Table 1: 3D pose estimation results on Human3.6M (protocol P1).

| Method | Type of images | 2D source size | 3D source size | Error (mm) |
|--------|---------------|---------------|---------------|-----------|
| Reg. | Real | 17,000 | 17,000 | 112.9 |
| Class. | Real | 17,000 | 17,000 | 149.7 |
| Reg. | Synth | 17,000 | 190,000 | 101.9 |
| Class. | Synth | 17,000 | 190,000 | 97.2 |
| Reg. | Real | 190,000 | 190,000 | 139.6 |
| Class. | Real | 190,000 | 190,000 | 97.7 |
| Reg. | Synth + Real | 207,000 | 190,000 | 125.5 |
| Class. | Synth + Real | 207,000 | 190,000 | **88.1** |

Table 2: Comparison with state-of-the-art results on Human3.6M. The average 3D pose error (mm) is reported before (Abs.) and after rigid 3D alignment for 2 different protocols. See text for details.

| Method | Abs. Error (P1) | Error (P1) | Abs. Error (P2) | Error (P2) |
|--------|----------------|-----------|----------------|-----------|
| Bo&Sminchisescu [5] | - | 117.9 | - | - |
| Kostrikov&Gall [18] | - | 115.7 | - | - |
| Iqbal et al. [42] | - | 108.3 | - | - |
| Li et al. [20] | - | - | 121.31 | - |
| Tekin et al. [35] | - | - | 124.97 | - |
| Zhou et al. [44] | - | - | **113.01** | - |
| Ours | 126 | **88.1** | 121.2 | 87.3 |

together. We observe that the classifier has similar performance when trained on synthetic or real images, which means that our image-based rendering engine synthesizes useful data. Furthermore, we can see that the classifier performs much better when trained on synthetic and real images together. This means that our data is different from the original data and allows the classifier to learn better features. Note that we retrain Alexnet from scratch. We found that it performed better than just fine-tuning a model pre-trained on Imagenet (3D error of 88.1mm vs 98.3mm with fine-tuning).

In Tab. 2, we compare our results to state-of-the-art approaches. We also report results for a second protocol (P2) employed in [20, 44, 35] where all the frames from subjects S9 and S11 are used for testing and only S1, S5, S6, S7 and S8 are used for training. Our best classifier, trained with a combination of synthetic and real data, outperforms state-of-the-art results in terms of 3D pose estimation for single frames. Zhou et al. [44] report better performance, but they integrate temporal information. Note that our method estimates absolute pose (including orientation w.r.t. the camera), which is not the case for other methods such as Bo et al. [5], who estimate a relative pose and do not provide 3D orientation.

## 4.2 Evaluation on Leeds Sport Dataset (LSP)

We now train our pose classifier using different combinations of training sources and use them to estimate 3D poses on images captured in-the-wild, i.e., LSP. Since 3D pose evaluation is not possible on this dataset, we instead compare 2D pose errors expressed in pixels and measure this error on the normalized $220 \times 220$ images following [44]. We compute the average 2D pose error over the 13 joints on both LSP and H3.6M (see Table 3).

As expected, we observe that when using a pool of the in-the-wild images to generate the synthetic data, the performance increases on LSP and drops on H3.6M, showing the importance of realistic images for good performance in-the-wild and the lack of generability of models trained on constrained indoor images. The error slightly increases in both cases when using the same number (190,000) of CMU 3D poses. The same drop was observed by [42] and can be explained by the fact that by CMU data covers a larger portions of the 3D pose space, resulting in a worse fit. The results improve on both test sets when considering more poses and synthetic images (2 millions). The larger drop in Abs 3D error and 2D error compared to 3D error means that a better camera view is estimated when using more synthetic data. In all cases, the performance (in pixel) is lower on LSP than on H3.6M due to the fact that the poses observed in LSP are more different from the ones in the CMU MoCap data. In Fig. 4 , we visualize the 2D pose error on LSP and Human3.6M 1) for different pools of annotated 2D images, 2) varying the number of synthesized training images and 3) considering different number of pose classes K. As expected using a bigger set of annotated images improves the

Table 3: Pose error on LSP and H3.6M using different sources for rendering the synthetic images.

| 2D source | 3D source | Num. of 3D poses | H3.6M Abs Error (mm) | H3.6M Error (mm) | H3.6M Error (pix) | LSP Error (pix) |
|---|---|---|---|---|---|---|
| H3.6M | H3.6M | 190,000 | 130.1 | 97.2 | 8.8 | 31.1 |
| MPII+LSP | H3.6M | 190,000 | 248.9 | 122.1 | 17.3 | 20.7 |
| MPII+LSP | CMU | 190,000 | 320.0 | 150.6 | 19.7 | 22.4 |
| MPII+LSP | CMU | $2.10^6$ | 216.5 | 138.0 | 11.2 | 13.8 |

Table 4: State-of-the-art results on LSP (2D pose error in pixels on normalized $220 \times 220$ images).

| Method | Feet | Knees | Hips | Hands | Elbows | Shoulder | Head | All |
|---|---|---|---|---|---|---|---|---|
| Wei et al. [39] | 6.6 | 5.3 | 4.8 | 8.6 | 7.0 | 5.2 | 5.3 | **6.2** |
| Pishchulin et al. [24] | 10.0 | 6.8 | 5.0 | 11.1 | 8.2 | 5.7 | 5.9 | 7.6 |
| Chen & Yuille [7] | 15.7 | 11.5 | 8.1 | 15.6 | 12.1 | 8.6 | 6.8 | 11.5 |
| Yang et al. [41] | 15.5 | 11.5 | 8.0 | 14.7 | 12.2 | 8.9 | 7.4 | 11.5 |
| Ours (Alexnet) | 19.1 | 13 | 4.9 | 21.4 | 16.6 | 10.5 | 10.3 | 13.8 |
| Ours (VGG) | 16.2 | 10.6 | 4.1 | 17.7 | 13.0 | 8.4 | 9.8 | 11.5 |

performance in-the-wild. Pose error converges both on LSP and H3.6M when using 1.5 million of images; using more than $K = 5000$ classes does not further improve the performance.

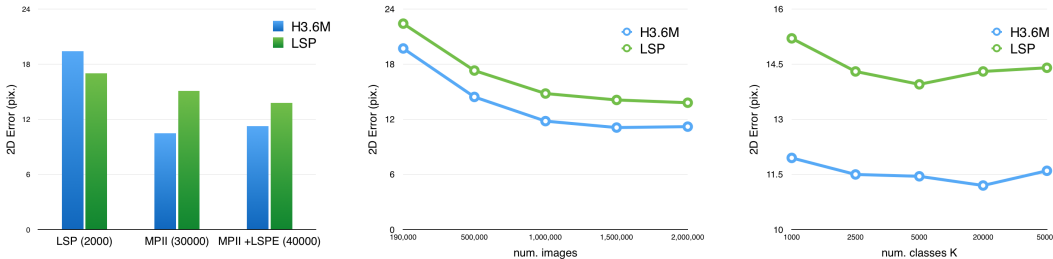

Figure 4: 2D pose error on LSP and Human3.6M using different pools of annotated images to generate 2 million of synthetic training images (left), varying the number of synthetic training images (center) and considering different number of pose classes K (right).

To further improve the performance, we also experiment with fine-tuning a VGG-16 architecture [33] for pose classification. By doing so, the average (normalized) 2D pose error decreases by 2.3 pixels. In Table 4, we compare our results on LSP to the state-of-the-art 2D pose estimation methods. Although our approach is designed to estimate a coarse 3D pose, its performances is comparable to recent 2D pose estimation methods [7, 41].

The qualitative results in Fig. 5 show that our algorithm correctly estimates the global 3D pose. After a visual analysis of the results, we found that failures occur in two cases: 1) when the observed pose does not belong to the MoCap training database, which is a limitation of purely holistic approaches, or 2) when there is a possible right-left or front-back confusion. We observed that this later case is often correct for subsequent top-scoring poses. This highlights a property of our approach that can keep multiple pose hypotheses which could be rescored adequately, for instance, using temporal information in videos.

## 5   Conclusion

In this paper, we introduce an approach for creating a synthetic training dataset of "in-the-wild" images and their corresponding 3D pose. Our algorithm artificially augments a dataset of real images with new synthetic images showing new poses and, importantly, with 3D pose annotations. We show that CNNs can be trained on artificial images and generalize well to real images. We train an end-to-end CNN classifier for 3D pose estimation and show that, with our synthetic training images, our method outperforms state-of-the-art results in terms of 3D pose estimation in controlled environments and shows promising results for in-the-wild images (LSP). In this paper, we have

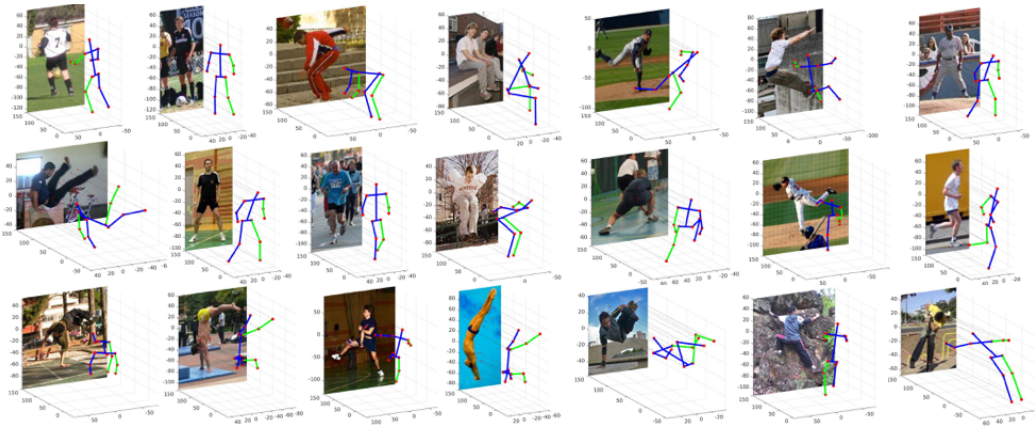

Figure 5: Qualitative results on LSP. We show correct 3D pose estimations (top 2 rows) and typical failure cases (bottom row) corresponding to unseen poses or right-left and front-back confusions.

estimated a coarse 3D pose by returning the average pose of the top scoring cluster. In future work, we will investigate how top scoring classes could be re-ranked and also how the pose could be refined.

**Acknowledgments.**    This work was supported by the European Commission under FP7 Marie Curie IOF grant (PIOF-GA-2012-328288) and partially supported by ERC advanced grant Allegro. We acknowledge the support of NVIDIA with the donation of the GPUs used for this research. We thank P. Weinzaepfel for his help and the anonymous reviewers for their comments and suggestions.

## Footnotes

[1] In practice, we do not search for occluded joints.

[2]http://mocap.cs.cmu.edu

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
