[Reviews · NeurIPS 2016]

Reviewer 1

Summary

This papers belongs to the current trend of producing synthetic databases as a proxi for real data when training CNNs. An interesting twist here is the use of MoCap data to guide the production of synthetic data and hence integrate existing, small scale collections. The procedure is well described and appears to be well thought of; results are strong.

Qualitative Assessment

I enjoyed reading this paper and I consider the research thread of trading real data with sysnthetic one important, so I think this would mae a strong poster. I have one question regarding the architecture chosen: you report using an AlexNet, where you only adapted the final layer. Did you try changing the convolutional layers? did you try any inception-type architecture? these last kind of architectures seem to work better on sysnthetic data than AlexNet (see for instance ModeNet). In general, further investigation on the optimal architecture would further increase the value of the work.

Confidence in this Review

2-Confident (read it all; understood it all reasonably well)


Reviewer 2

Summary

Authors propose a method to generate training samples for the task of 3D pose estimation from single monocular images. To this end they utilize 2 data sources: 3D motion capture data and images annotated with 2D poses. For a given 3D pose and a camera viewpoint they retrieve N images (where N is number of joints), which have the most similar local geometry around respective joints as the 2D projection of the source 3D pose. Then, they stitch the N images into a coherent one with pose aware smoothing. The resulting synthetic dataset is added to the existing dataset which is annotated with 3D poses. 3D poses are clustered into K=5000 pose classes and a holistic CNN is trained for a classification task. At test time 3D pose is estimated as average pose of top scoring classes. This method achieves state-of-the-art performance on Human3.6M dataset.

Qualitative Assessment

- Technical quality. Authors compare their method to a very recent state-of-the-art method and use the same evaluation protocol. They present detailed evaluation of how synthetically generated data fairs against real data and how using them both together improves performance. Additionally authors demonstrate the advantage of formulating 3D pose estimation as a classification problem over direct regression to joint coordinates. This is facilitated by a large number of training samples, which they show is crucial for the classification CNN: in the small data regime full body regression actually outperforms 3D pose classification. However, authors only evaluate their method using evaluation protocol described in [17, 40]. It would be nice to show how method compares to [18, 19] which use a different protocol on Human3.6M, even though it is an easier problem according to [17]. - Novelty/Originality To the best of my knowledge this work is the first to synthesize training samples from real world images for the task of human pose estimation. This is an alternative approach to rendering images given the 3D pose of a subject. Despite the fact that generated images do not match real-world images in realism, using them is beneficial. - Potential impact or usefulness In order to train Deep Neural Networks large datasets are required, and this work explores how to generate such training sets. Therefore it can potentially be interesting to the community. Since the method allows to generate diverse samples for 2D pose estimation as well, showing results on 2D pose estimation would demonstrate whether the proposed approach is applicable to other problems. Thus an experiment with augmenting typical 2D pose training sets such as MPII that shows an improvement in 2D pose estimation would be of great value. Otherwise, quantitative evaluation on only one dataset slightly questions the generality of the approach. The evaluation on LSP while interesting, does not give insight of how the proposed data augmentation method can push the 2D pose estimation, because it does not use an established evaluation metric. - Clarity and presentation The data generation method is described with great detail and I had no problem understanding it. The paper reads well overall, however there are a number of spelling mistakes to be corrected. - Final assessment Overall, I like the paper. Authors clearly demonstrate that adding training samples, generated with their method, improves performance of 3D pose estimation using a classification-based CNN. However I feel that quantitative evaluation on just one benchmark is not enough to conclude on generality of the data generation method, especially because generated images do not appear entirely realistic.

Confidence in this Review

2-Confident (read it all; understood it all reasonably well)


Reviewer 3

Summary

The paper proposes a process for generating realistically-looking synthetic data for 3D human pose estimation, given a dataset of images with 2D poses in the wild and a dataset of motion capture 3D poses. The main contribution is an algorithm that performs mosaicing of fractions of the images from the 2D dataset and performs pose-aware blending. Visually these synthetic examples are interesting. The paper then evaluates 3D and 2D pose classification with and without this additional synthetic training data on the Human-3.6M and the LSP datasets.

Qualitative Assessment

The generation process is novel and interesting and the results promising, but there is no reason for this to add value for predicting the depth dimension of the pose. It is likely that the improvement in results by using synthetic data comes from better estimation of the image-plane dimensions and this shows for example in table 3, e.g. error in H3.6M drops in one case from 150.6 to 130 mm in 3D but in pixels drops much more significantly from 19.7 to 10.5 pixels. The approach would be more usefully geared towards improving 2D pose estimation in the wild, where data is also still scarce. Another thing that can be improved is in the 2D results using standard evaluation measures such as pck, in order to be able to compare with the state-of-the-art.

Confidence in this Review

2-Confident (read it all; understood it all reasonably well)


Reviewer 4

Summary

This paper proposes a simulation based approach to estimate 3D human body poses from 2D images in the wild. There are a few issues as below: * The title is a bit misleading: The experiments are not restricted to be in the wild. In fact, the majority of the data considered such as H3.6M and MPII and CMU Mocap are controlled indoor scenarios. Only the LSP dataset is outdoor. The paper can instead focus on only in the wild (i.e. outdoor) situations to best match with the current title. * The simulated images are not photo-realistic, as e.g. illustrated in Fig.1. The authors argue that this may be OK for practical applications. Still it is questionable, since the global contextual information including both foregrounds and backgrounds are not preserved in the simulated data (e.g. Fig.2). * In general, 3D pose estimation is a regression problem rather than a classification problem, as naturally with the space of 3D human pose articulation. It is a bit surprise that it is turned into a classification problem with K=5000 pose classes. This change of problem is quite fundamental and may lead to important consequences. * Eq.3 seems to be wrong: should it be min instead of max here, as we are after the nearest example that minimizing the matching error here?

Qualitative Assessment

see above

Confidence in this Review

2-Confident (read it all; understood it all reasonably well)


Reviewer 5

Summary

The paper proposed a solution to generate a large set of photorealistic synthetic images of humans with 3D pose annotations. The main idea is to use an image-based synthesis engine that artificially augments a dataset of real images with 2D human pose annotations using 3D Motion Capture (MoCap) data. Given a candidate 3D pose, the proposed algorithm selects for each joint an image whose 2D pose locally matches the projected 3D pose. The selected images are then combined to generate a new synthetic image by stitching local image patches in a kinematically constrained manner.

Qualitative Assessment

This paper does deal with an important problem, lacking of training data of 2D images of human annotated with 3D poses, especially for CNN pipelines. But I think that this problem can be easily solved with image warping works in computer graphics fields. That is, data augmentation with real images, only warping human image region with motion capture data. Pls refer the the following paper. Character Animation from 2D Pictures and 3D Motion Data Alexander Hornung, Ellen Dekkers, Leif Kobbelt ACM Transactions on Graphics, 2007, vol. 26(1) The vividness of synthesized images is important for training, since such data can provide similar data property to real test images, and then lead to good testing performance. However,synthesized images in the proposed method(See Fig 2 (e)) is far from vivid. The image warping works in computer graphics fields can provide better results, and should be a better solution for data augmentation.

Confidence in this Review

2-Confident (read it all; understood it all reasonably well)


Reviewer 6

Summary

The authors suggest a novel method to synthesize training data for 3d pose estimation from 2D images. The novelty resides in the fact that synthesized training data is 2D stiched in an intelligent way, instead of the commonly used 3d rendering. The resulting images look artificial in a different way than rendered images, but the authors show that this strange aspect actually help the 3d pose estimator generalizing better.

Qualitative Assessment

The paper is clear, and method original and sound. If only quite "application limited". However it showcases that CNNs can be trained on artificially looking images and still generalize well, which I think its important and novel. As a limitation, I have the feeling that the impact of occlusions have not been discussed sufficiently. Also tables are quite confusing, not stating what is training and what is testing. l.217 faire -> fair l.219 (see Table 3) or (see results in Table 3) l.223 slight -> slightly

Confidence in this Review

2-Confident (read it all; understood it all reasonably well)